# Planned vs. Performed Treatment Regimens in Diabetic Macular Edema: Real-World Evidence from the PACIFIC Study

**DOI:** 10.3390/jcm14093120

**Published:** 2025-04-30

**Authors:** Christos Haritoglou, Matthias Iwersen, Bettina Müller, Erik Beeke, Hüsnü Berk, Matthias Grüb, Katrin Lorenz, Martin Scheffler, Focke Ziemssen

**Affiliations:** 1Augenklinik Herzog Carl Theodor, Nymphenburger Str. 43, 80335 München, Germany; 2Novartis Pharma GmbH, Roonstr. 25, 90429 Nürnberg, Germanybettina.mueller@novartis.com (B.M.); 3Visualeins MVZ für Augenheilkunde und Anästhesie GmbH, Am Finkenhügel 7 B, 49076 Osnabrück, Germany; 4St. Elisabeth-Krankenhaus GmbH, Klinik für Augenheilkunde, Werthmannstr. 1, 50935 Köln, Germany; 5Matthias Grüb & Kollegen, Bahnhofstr. 7-9, 79206 Breisach am Rhein, Germany; matthias.grueb@grueb.eu; 6Augenklinik und Poliklinik der Universitätsmedizin Mainz, Langenbeckstr. 1, 55131 Mainz, Germany; 7Augenheilkunde Rhauderfehn, Dr. med. Martin Scheffler, Rhauderwieke 3, 26817 Rhauderfehn, Germany; 8Klinik für Augenheilkunde, Universitätsklinikum Leipzig, Liebigstraße 12, 04103 Leipzig, Germany; 9Department für Augenheilkunde, Eberhard-Karls-Universität Tübingen, Elfriede-Aulhorn-Str. 7, 72076 Tübingen, Germany

**Keywords:** diabetic macular edema (DME), vascular endothelial growth factor (VEGF) inhibitors, real-world evidence, treatment adherence, intravitreal injections, observational study, treatment strategies, clinical guidelines, treatment deviations, patient-centered care

## Abstract

**Background:** Intravitreal injections of vascular endothelial growth factor (VEGF) inhibitors are standard for diabetic macular edema (DME), yet a gap exists between clinical guidelines and actual practices. This study aimed to investigate the extent of deviation between physician-planned and actually performed treatment regimens. **Methods:** The PACIFIC study (NCT04847895) was a prospective, multicenter, non-interventional study conducted in Germany, the Netherlands, and Switzerland. A total of 910 patients with DME receiving ranibizumab were enrolled. Physicians documented the intended treatment regimen at baseline, and actual treatment patterns were retrospectively derived from the timing of visits and injections over a 24-month observation period. **Results:** Although most physicians initially planned fixed or pro re nata (PRN) regimens, 77% of pretreated and 73% of treatment-naïve patients ultimately followed a monitor and extend strategy. Treatment discontinuation was frequent (58.8% and 59.4%, respectively), and injection frequencies remained below recommended levels, although central retinal thickness improved over time. **Conclusions:** The study highlights a consistent and clinically relevant discrepancy between planned and actual treatment delivery in DME care, underscoring the need for better adherence to guideline-informed strategies in routine practice.

## 1. Introduction

Intravitreal injections of vascular endothelial growth factor (VEGF) inhibitors, such as ranibizumab, represent the standard of care for diabetic macular edema (DME). Various clinical trials have demonstrated the efficacy and safety of ranibizumab in treating DME [1,2,3,4]. In addition, researchers have published several studies demonstrating the effectiveness of ranibizumab in treating DME in a real-world setting. However, these studies have methodological limitations; for instance, previous studies did not compare the initially planned treatment regimen to the one actually performed [5,6,7,8]. In particular, the number of treatments was low in previous retrospective cohorts. According to the observational study design, there were several possible explanations: either only a few injections were initially agreed upon, a conscious or unconscious decision was made against further injections, or early withdrawal from longer follow-up distorted the figures.

Since clinicians often observe continuous but slow improvement under initial treatment and macular fluid needs time to decline [9], guidelines recommend intensive treatment, e.g., providing an initial series of injections, followed by further injections as needed (“pro re nata”) until maximum visual acuity is achieved (German guidelines [10]). Retina specialists consider the condition to be stable when the patient’s visual acuity remains unchanged and/or no signs of disease activity are detected during consecutive monthly tests. They monitor these parameters by assessing best corrected visual acuity (BCVA) and using optical coherence tomography (OCT) [11]. The initial intensity has been shown to be important for later outcomes [12,13,14]. After the first 6 months, a “treat and extend” regimen might be an excellent alternative to the “pro re nata” regimen, particularly if only one eye is affected [10].

The poorer outcome of crossover groups with late treatment started proved the impact on therapy outcomes [15]. Visual acuity at the start of therapy is an important prognostic factor for the level of visual acuity that can be reached by treatment [16,17,18,19,20,21]. Clearly, relatively good visual acuity at baseline limits “room for improvement” in terms of gains in letters. However, despite lower relative gains in letters, better absolute visual acuity can be achieved in these patients, and high visual acuity can be maintained in comparison to starting treatment with low visual acuity. Thus, the aim should be to diagnose DME as early as possible and initiate treatment immediately [22].

At the time of planning the PACIFIC study, no data regarding the efficacy of different treatment regimens used in routine clinical practice had been available. Thus, the PACIFIC study was conducted to determine the utilization and effectiveness of these various treatment regimens in routine clinical practice and their dependence on the application of different treatment regimens.

## 2. Materials and Methods

### 2.1. Study Design

The PACIFIC study was an observational, multicenter, open-label, single-arm study of patients with DME in Germany, the Netherlands, and Switzerland who were treated with ranibizumab.

Patient enrollment started in June 2015 and ended in March 2019. A minimum of one follow-up visit per year was required to maintain patient participation in the study over the prospective observational period of 24 months. This was a non-interventional study (NIS) in which no therapy, diagnostic/therapeutic procedure, or visit schedule was imposed.

### 2.2. Subjects

Patients could be included in the study if they were being treated with ranibizumab for any approved indication and gave written informed consent. Treatment-naïve (untreated) and pretreated patients were included. Pretreatment was defined as prior treatment with ranibizumab, other anti-VEGF drugs, intravitreal corticosteroids, photodynamic therapy, or laser coagulation before the start of the PACIFIC study (baseline). A total of 185 sites participated in the PACIFIC study, with 4948 patients in the safety evaluation set (SES) and 4932 patients in the full analysis set (FAS) for all indications. Of these, 913 patients in the SES cohort and 910 patients in the FAS had DME and were the subject of the present evaluation.

### 2.3. Statistical Methods

In the present analysis, descriptive summary statistics were calculated for the data from all the DME patients. Patient characteristics were documented at baseline. Variables such as details of ranibizumab injections and planned treatment regimens, BCVA examinations, OCT examinations, and adverse events (AEs) were documented at each visit. Diseases from medical history and AEs were coded using the Medical Dictionary for Regulatory Activities (MedDRA) version 24.0. For variables with predictive validity, appropriate strata were created. Statistics were calculated using the SAS software version 9.4 (SAS Institute, Cary, NC, USA). All safety-relevant analyses were performed for the SES, and all analyses relevant for utilization and effectiveness were performed for the FAS. Analyses were performed according to a predefined analysis plan.

The intended treatment regimens were documented by the treating physicians. The actual treatment regimens used were derived from the documented time points. Treatment regimens were defined as follows: “fixed” scheme, with regular intervals between follow-up visits and reinjection at every visit; “pro re nata”, with regular intervals between follow-up visits and reinjection as needed; “treat and extend”, with intervals between follow-up visits extended step by step and reinjection at every visit; and “monitor and extend”, with variable intervals between follow-up visits and reinjection as needed.

## 3. Results

### 3.1. Participants

A total of 913 patients with DME (in the SES) were enrolled in the PACIFIC study, and 910 patients were included in the FAS. Among those, 458 patients had previously received treatment, and 452 patients were treatment naïve. Most patients were from Germany (423 pretreated, 433 naïve), and few were from the Netherlands (19 pretreated, 0 naïve) or Switzerland (16 pretreated, 19 naïve). Among patients from Germany, the vast majority had public health insurance (92.0% of pretreated patients, 93.3% of treatment-naïve patients).

Demographics and baseline disease characteristics are shown in Table 1. Slightly more patients in the PACIFIC study were male, and the mean age was approximately 66 years for both pretreated and treatment-naïve patients. The mean baseline HbA1c value was approximately 52 mmol/mol for pretreated patients and 49 mmol/mol for treatment-naïve patients. The mean baseline BCVA, measured as early treatment diabetic retinopathy study (ETDRS) letters, was 66.3 of the ETDRS letters for pretreated patients compared to 64.1 of the ETDRS letters for treatment-naïve patients. Most patients had a baseline OCT examination, with a mean central retinal thickness of 330.0 µm for pretreated patients and 352.8 µm for treatment-naïve patients. The documented fundus lesions did not differ between the naïve and pretreated study participants (Appendix A, Table A1).

### 3.2. Treatments

Pretreated patients received a mean (±SD) number of 5.8 ± 2.9 injections (*n* = 266) of ranibizumab for the study eye in the first year (among those patients with documentation for at least the first year) and 10.6 ± 5.4 injections (*n* = 55) in two years (among those patients with documentation for at least two years). Treatment-naïve patients with DME received 6.4 ± 2.8 injections (*n* = 264) in the first year and 9.2 ± 4.5 injections (*n* = 52) in two years.

In the first 6 months, the pretreated patients had a mean (±SD) of 3.5 ± 1.6 injections. In the first 6 months, treatment-naïve patients had a mean (±SD) of 4.1 ± 1.5 injections.

The median time between the baseline BCVA examination and the first ranibizumab injection was 6 days (mean ± SD: 12.6 ± 17.1 days; *n* = 386) for pretreated patients with DME and 5 days (mean ± SD: 10.7 ± 15.1 days; *n* = 408) for treatment-naïve patients. The median time between subsequent study eye injections was 44.4 days (mean ± SD: 50.8 ± 23.1 days; *n* = 398) in pretreated patients and 39.2 days (mean ± SD: 45.1 ± 19.9 days; *n* = 432) in treatment-naïve patients.

The course of concomitant laser treatments over the observational period for treatment-naïve patients is shown in Figure 1. Concomitant laser treatments for pretreated patients are provided in Appendix A, Figure A1, and a cumulative depiction of treatment-naïve and pretreated patients is provided in Appendix A, Figure A2. Most patients who received concomitant laser treatments had at least one laser treatment within the first 6 months. In total, 9.0% of the pretreated patients and 15.1% of the treatment-naïve patients received concomitant laser treatments during the observational period.

The pretreated patients remained in the documented study treatment for a median follow-up of 442 days, i.e., approximately 14.5 months, and treatment-naïve patients for a median of 453 days, i.e., approximately 15 months. Premature discontinuation of the study was documented for 258 of the 439 pretreated patients (58.8%) and for 252 of the 424 treatment-naïve patients (59.4%). The percentage of patients who prematurely discontinued the study was slightly greater among patients with renal insufficiency (68.0% of 25 pretreated patients, 88.9% of 18 treatment-naïve patients) and among patients with HbA1c >9% (66.7% of 21 pretreated patients, 68.4% of 19 treatment-naïve patients).

### 3.3. Treatment Regimens

The intended treatment regimens, as documented at baseline, as well as the actually performed treatment regimens in the last month, based on statistical derivation from temporal patterns of visits and injections, are shown in Figure 2. At baseline, a “fixed” scheme and “pro re nata” were the most commonly used treatment regimens. For pretreated patients, “pro re nata” was slightly more frequent (41% “pro re nata”, 34% “fixed”), while for treatment-naïve patients, a “fixed” scheme was more frequent (46% “fixed”, 35% “pro re nata”). However, overall, the treatment regimen used was “monitor and extend” for the majority of patients (77% of pretreated patients and 73% of treatment-naïve patients). The time-dependent trends in treatment regimens over 24 months are shown in Appendix A, Figure A3 and Figure A4 for pretreated and treatment-naïve patients, respectively.

### 3.4. Visual Acuity

At the end of the observational period (month 24), the mean (±SD) BCVA was logMAR 0.270 ± 0.274 or 71.5 ± 13.7 of ETDRS letters (n = 71) for pretreated patients and logMAR 0.353 ± 0.330 or 67.3 ± 16.5 of ETDRS letters (n = 76) for treatment-naïve patients. Compared to the baseline BCVA, the BCVA of the study eye improved during each month of the study. The changes from baseline for pretreated and treatment-naïve patients are shown in Figure 3.

Pretreated patients received a total mean (±SD) number of 9.2 ± 5.2 OCT examinations of the study eye (n = 55), and treatment-naïve patients received 8.6 ± 5.7 OCT examinations (n = 52, among patients with documentation for at least two years). Decreases in the central retinal thickness of the study eye were observed at all visits compared to baseline. The course of central retinal thickness by study month is provided in Appendix A, Figure A5 for pretreated patients and in Appendix A, Figure A6 for treatment-naïve patients.

### 3.5. Adverse Events

Among pretreated patients, 41.2% (of 459 patients in the SES) experienced AEs. Severe AEs (SAEs) were reported for 27.7%, and serious adverse drug reactions (SADRs) were reported for 6.1%. The most frequently reported AEs were diabetic retinal edema (5.0%), cystoid macular edema (4.6%), retinal edema (4.1%), and concomitant disease aggravation (4.1%). Of a total of 551 AEs, 33.6% had mild intensity, 40.1% had moderate intensity, and 16.3% had severe intensity. The incidence of AEs was greater among pretreated patients with an HbA1c >9% (66.7% of 21 patients).

Among treatment-naïve patients, 33.5% (of 454 patients in the SES) experienced AEs. SAEs were reported for 22.5%, and SADRs were reported for 5.3%. The most frequently reported AEs were diabetic retinal edema (5.1%), cystoid macular edema (4.0%), concomitant aggravated disease (3.3%), and reduced visual acuity (3.1%). Of a total of 438 AEs, 33.3% had mild intensity, 40.2% had moderate intensity, and 16.9% had severe intensity.

## 4. Discussion

The PACIFIC study provides real-life data on therapy with ranibizumab in local routine clinical practice. It is noteworthy that this study was initiated before the latest publication of the German guidelines [10], which recommended treatment initiation by six monthly intravitreal injections of a VEGF inhibitor. Although the corresponding study data should be carefully interpreted due to certain limitations (selection bias, loss of documentation), five important observations can be made as conclusions about everyday treatment:(1)Over the last few years, a learning curve has been recognized, which is reflected in key parameters. The median treatment delay was 6 days for pretreated patients and 5 days for treatment-naïve patients in this study. This is notably shorter than in the preceding OCEAN study (conducted between 2011 and 2016), which reported a median treatment delay of approximately 21 days for DME patients [23]. This finding indicates an increased awareness of the importance of early treatment among ophthalmologists in recent years. Success and long-term preservation of visual function depend largely on the function at the start of treatment [24,25];(2)The intended treatment pattern showed notable differences compared to the actually performed regimen. At baseline, a “fixed” scheme and “pro re nata” were the most commonly used treatment regimens. This would have been in accordance with the DOG guidelines, which recommend a “fixed” scheme for the first 6 months, followed by “pro re nata” treatment [10]. However, in reality, these regimens were only performed for a small percentage of patients. This finding indicates that undertreatment is still an issue for DME patients [26]. Over the course of the observational period, a shift toward a “monitor and extend” regimen was observed based on the statistical derivation. It is possible that due to the study’s defined query and temporal limits, the data led to an interpretation of the study as a “monitor and extend” regimen in many cases, even if not intended by the physician. It seems important to look for unconscious influences and other relevant factors [27]. In the future, attention should be given to realistic agreements to improve the quality of educational discussions and provide informed consent;(3)Concomitant laser treatment has much less significance than described in clinical studies. Approximately 41–64% of treatment-naïve DME patients (DRCR.net Protocol T from 2012 to 2014) received at least one laser treatment over 2 years, and 23% of pretreated patients and 15% of treatment-naïve patients among the PACIFIC participants underwent concomitant laser treatment. However, it remains unclear whether this difference is ultimately due to the non-interventional nature of the study or to the basic philosophy of a less aggressive combined approach in Europe vs. the USA [28]. Presumably, both influences are likely to play a role if the decision to use a focal-grid laser in addition to anti-VEGF drugs does not follow a systematic algorithm. Although the stability of the achieved visual acuity gain is more relevant for retreatment than retinal morphology is, given the proportions of persistent fluid in DME [11], the strategy may not always be effective because of unclear expectations and perceived non-response. Although combined laser treatment did not show a general benefit in studies, there is indirect evidence for the benefits of targeted supplementary laser treatment over time [29];(4)The trend toward low treatment numbers was similar between naïve and pretreated patients. Pretreated patients in the PACIFIC study received a mean of 10.6 injections, and treatment-naïve patients received 9.2 injections (among those patients with documentation for at least two years). Thus, patients received notably more injections than in the previous observational OCEAN study (5.5 injections) [30], but still fewer injections than in clinical trials [31,32];(5)Treatment discontinuation is still a relevant problem in this treatment indication [33]. Aggravated general conditions often cause patients to discontinue treatment. In the PACIFIC study, patients with very high HbA1c values (>9%) were more likely to be discontinued prematurely and experience more AEs. There were no major outliers in terms of insurance or race, but the analyzed cohort was quite homogeneous and of Central European ancestry [34].

The efficacy of the treatment was assessed in representative cohorts. However, by providing real-life data on DME therapy, observational insights from routine clinical practice may lead to a better understanding of the status quo and challenges in patient health care in real-life settings [35,36].

Earlier studies were unable to reveal the considerable deviations between original planning and actual implementation because the planning perspective was not recorded. Specifically, the findings raise the question of whether and to what extent practitioners are aware of the pathways to undertreatment and treatment discontinuation. Overt or hidden resistance, if not the justified and clearly articulated will of the patient, can be relevant. This makes it all the more important that the opportunities for better information—including the relevance of the necessary adherence—are utilized in the future [37].

## 5. Conclusions

In summary, compared with those of previous studies, the present study showed a slight improvement in the real-world treatment of DME patients with regard to treatment delay and the number of injections given during the first year of treatment. However, the current local recommendations [10], suggesting intensive treatment (e.g., comprising six monthly injections followed by further injections based on specific reinjection criteria), were not achieved.

Although regimens clearly define the sequence of control of morphological parameters, fixed re-treatment and determination of the interval for the ‘treat and extend‘ regime, the actual observed behavior after appropriate planning corresponded to completely different patterns. Various factors may contribute to the lack of regular re-treatment while extending, and patients interrupting or terminating the strategy despite the need for intensive treatment.

Deviations in the execution of the plans definitely occur. The search for the reasons and causes can help to improve the treatment of DME in different healthcare settings.

## Figures and Tables

**Figure 1 jcm-14-03120-f001:**
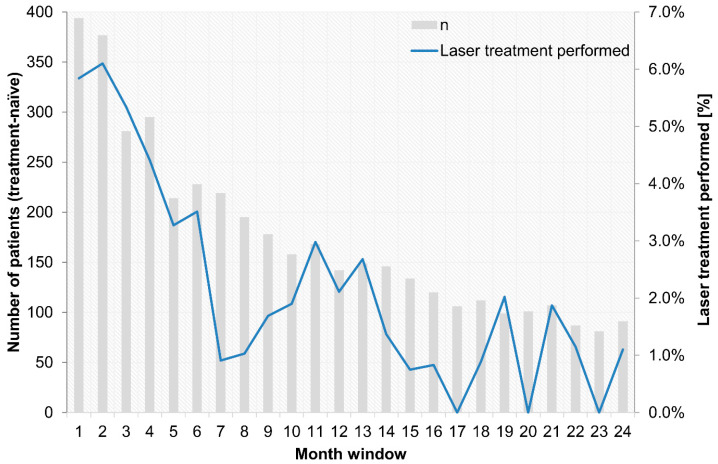
Concomitant laser treatment over the 24-month observational period for treatment-naïve patients.

**Figure 2 jcm-14-03120-f002:**
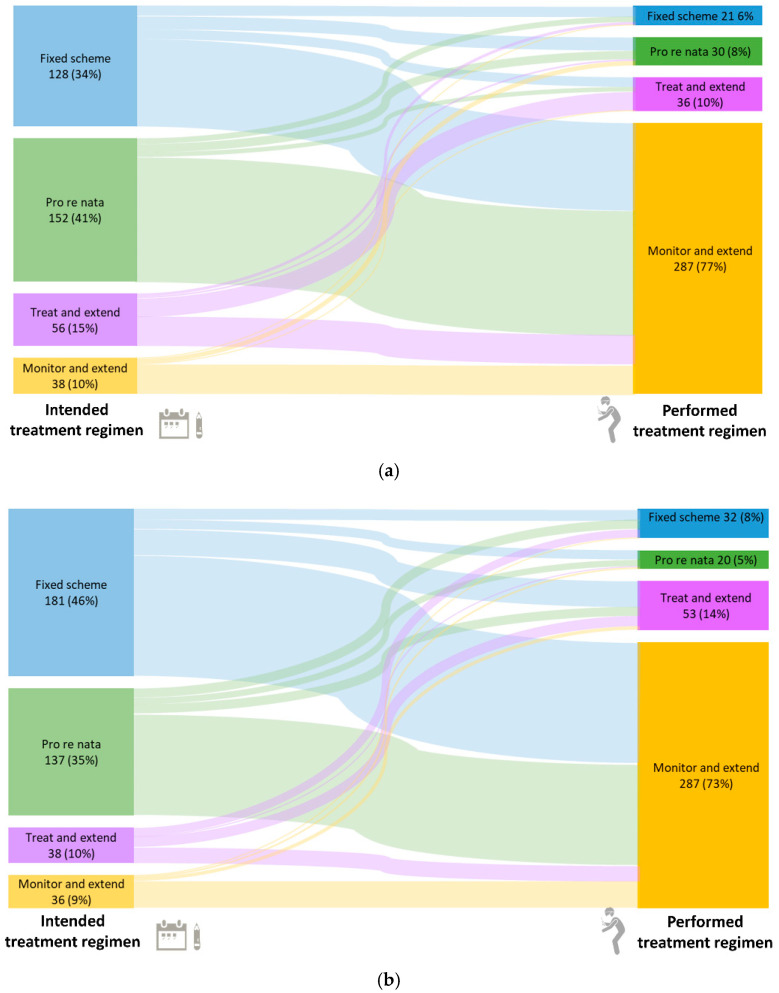
Intended treatment regimens at baseline and actual treatment regimens in the last month for pretreated (**a**) and treatment-naïve (**b**) patients. Percentages are based on non-missing observations. In total, 443 pretreated patients had observations in the last month. Information on the treatment regimen was missing for 26 patients, and for 43 patients, the intended treatment regimen was changed during the study; therefore, these patients are not included in this figure. Of the 443 naïve patients with observations in the last month, 16 patients were missing information, and the intended treatment regimen was changed for 35 patients.

**Figure 3 jcm-14-03120-f003:**
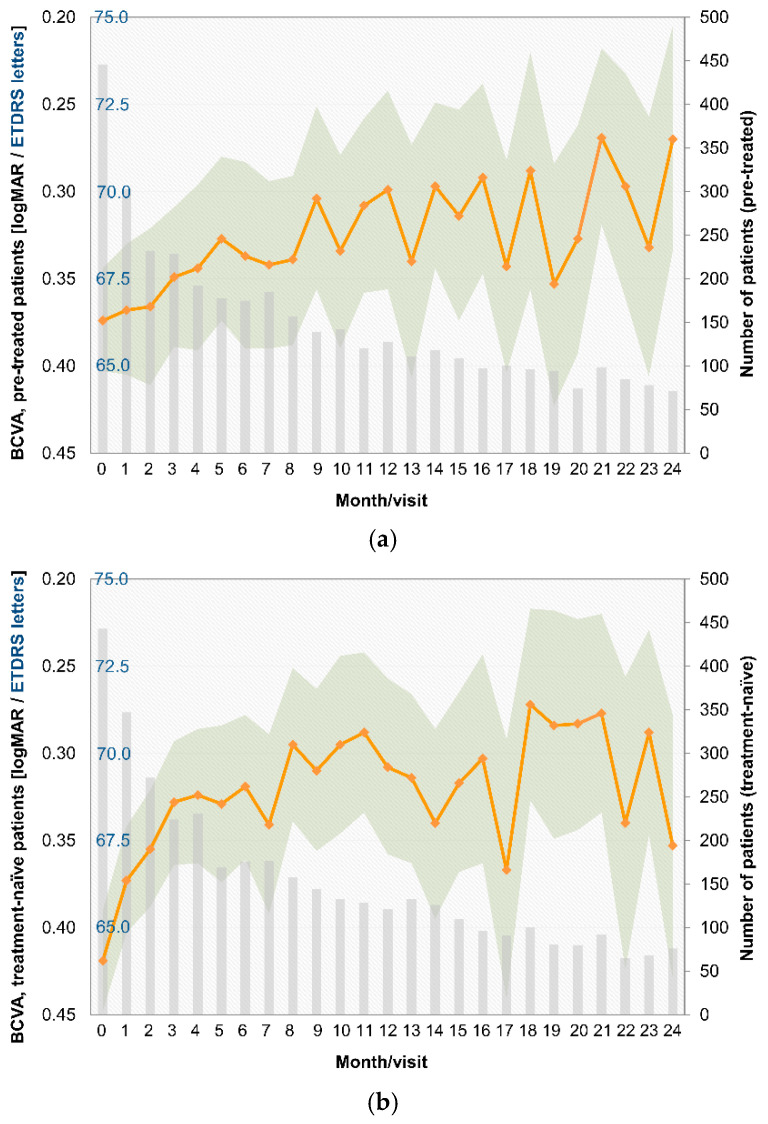
BCVA by month, logMAR, and ETDRS values for pretreated (**a**) and treatment-naïve patients (**b**).

**Table 1 jcm-14-03120-t001:** Demographics and baseline disease characteristics of patients with DME stratified by pretreatment status (FAS).

Parameter	Pretreated N = 458	Treatment-Naïve N = 452
Demographics		
Sex, n (%)		
Male	275 (60.0%)	267 (59.1%)
Female	183 (40.0%)	185 (40.9%)
Age at initial study visit [years], mean (SD)	66.4 (11.7)	66.3 (11.8)
Height [cm], mean (SD)	171.0 (9.1)	170.4 (9.3)
Weight [kg], mean (SD)	86.1 (18.2)	86.4 (17.1)
**Most commonly reported diseases from medical history, n (%)** ^a^		
Diabetes mellitus ^b^	456 (99.6%)	452 (100.0%)
Arterial hypertension ^b^	236 (51.5%)	238 (52.7%)
Hyperlipidemia ^b^	44 (9.6%)	30 (6.6%)
Apoplexy	20 (4.4%)	26 (5.8%)
Myocardial infarct	21 (4.6%)	22 (4.9%)
Renal insufficiency ^b^	25 (5.5%)	18 (4.0%)
**Patients with diabetes mellitus: specification of type, n (%)**		
n	456	452
Type I	50 (11.0%)	46 (10.2%)
Type II	358 (78.5%)	347 (76.8%)
Unknown/Missingc	48 (10.5%)	59 (13.1%)
**Diabetes mellitus: baseline HbA1c value**		
N	189	184
Mean (SD) [mmol/mol]	51.7 (25.7)	49.4 (26.1)
Mean (SD) [%]	6.9 (2.4)	6.7 (2.4)
**Baseline BCVA examination of study eye**		
n	446	443
logMAR, mean (SD)	0.374 (0.317)	0.419 (0.310)
ETDRS letters, mean (SD)	66.3 (15.9)	64.1 (15.5)
**Baseline OCT examination of study eye**		
**Baseline OCT examination performed?, n (%)**		
Yes	313 (68.3%)	379 (83.8%)
No	129 (28.2%)	71 (15.7%)
Unknown/Missing ^c^	16 (3.5%)	2 (0.4%)
**Baseline OCT: central retinal thickness [µm]**		
n	302	369
Mean (SD)	330.0 (108.1)	352.8 (114.1)

^a^ Multiple responses permitted. ^b^ Disease from medical history was explicitly documented as ongoing at baseline for the majority of patients. ^c^ Data were either reported as unknown or not reported (missing). DME: diabetic macular edema, FAS: full analysis set, HbA1c: hemoglobin A1c, N: number of patients in the analysis set, n: number of non-missing observations, SD: standard deviation.

## Data Availability

Patients provided signed informed consent for the publication of their data.

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
