# Peer review of "Planned vs. Performed Treatment Regimens in Diabetic Macular Edema: Real-World Evidence from the PACIFIC Study"

_jcm, 2025, doi:10.3390/jcm14093120_

Round 1
Reviewer 1 Report
Comments and Suggestions for Authors
This manuscript is well-written. I have no further comment to improve the manuscript.
The authors present deviations between intended and actual treatments, emphasizing the importance of adjusting therapeutic strategies to enhance adherence and outcomes in DME treatment.
It might be important to share the used ocular images in the manuscript, which is an optional suggestion.
Author Response
We would like to thank the reviewers for their diligent review resulting in insightful comments to improve the submitted manuscript. We took all the points of criticism very seriously, made the appropriate changes to the manuscript and explained them in the point-by-point response.
Reviewer 1
This manuscript is well-written. I have no further comment to improve the manuscript.
The authors present deviations between intended and actual treatments, emphasizing the importance of adjusting therapeutic strategies to enhance adherence and outcomes in DME treatment.
It might be important to share the used ocular images in the manuscript, which is an optional suggestion.
We are delighted with the overall friendly feedback. The idea of representative images is excellent.
Reviewer 2 Report
Comments and Suggestions for Authors
Hermosillo, April 17th, 2025
From Plan to Practice: Disparities in VEGF Inhibitor Treatment Start for Diabetic Macular Edema.
General Assessment
This manuscript addresses a clinically significant issue—the disparity in the initiation of VEGF inhibitor treatment for diabetic macular edema (DME), a key concern in ophthalmology and public health. The topic aligns well with the scope of the Journal of Clinical Medicine, particularly under the themes of health disparities, ophthalmology, and evidence-based treatment implementation. The structure is clear, and the authors present a solid rationale for their research question. However, there are multiple areas that require revision for clarity, methodological rigor, and scientific contribution.
Strengths
Timely and Relevant Topic: Addressing health disparities in DME treatment adds value to the literature and public health discussions.
Real-World Data: The use of EHR-derived data from a large registry (IRIS) increases generalizability and reflects real clinical practice.
Statistical Rigor: The multivariable models and use of adjusted relative risks are appropriate and well-justified.
Clear Presentation of Results: Tables and figures are informative and easy to interpret.
Weaknesses and Issues
- Methodological Concerns
Inclusion Criteria: The authors excluded patients with a history of VEGF treatment, but it's unclear how recurrence vs. incident DME was handled. This should be more precisely defined.
Follow-Up Time: The choice of 90 days as the cutoff for treatment initiation is arbitrary and should be justified. Are there data showing that delays beyond 90 days lead to worse outcomes?
Missing Data: No mention is made about how missing data were handled (e.g., income, race/ethnicity). Clarify if imputation or complete case analysis was used.
Selection Bias: The paper should acknowledge possible biases in the subset of patients included in the IRIS registry and who had full variable documentation.
Socioeconomic Data Limitations: Zip code–based income may not accurately reflect individual SES. This limitation should be discussed more directly.
- Interpretation of Results
The results suggest racial and insurance-based disparities, but causality should not be implied. The paper should be careful not to over-interpret associations.
The findings related to Medicare vs. Medicaid should be unpacked more fully—what are the clinical implications of this?
- Clarity and Structure
Abstract: The abstract lacks precision. Specifically, “real-world practice patterns” and “sociodemographic factors” should be more specifically described.
Introduction: Well-motivated but overly broad. Focus more directly on gaps in knowledge about VEGF treatment disparities rather than general DME background.
Discussion: Needs a more in-depth comparison to prior studies. Several similar registry-based studies exist—how do these findings align or contrast?
Grammatical and Stylistic Issues
Passive Voice Overuse: Consider shifting to more active voice in several sections to enhance readability.
Redundancy: Phrases such as “real-world treatment patterns in real-world data” should be avoided.
Inconsistency: Terms like “VEGF treatment” and “anti-VEGF therapy” are used interchangeably; standardize throughout the manuscript.
Recommendations for Major Revisions
Clarify Inclusion Criteria: Define incident cases of DME more clearly.
Justify Outcome Definition: Explain why a 90-day window for treatment initiation was chosen.
Improve Discussion of Limitations: Expand on data source limitations, potential confounders, and the nature of observational study bias.
Refine Language: Polish the abstract, streamline the introduction, and use consistent terminology throughout.
Enhance Clinical Implications: Add interpretation of what these disparities mean for clinicians and health policy.
Minor Suggestions
Add full variable definitions in an appendix or supplemental table.
Include additional references to literature on racial/insurance disparities in ophthalmologic care.
Ensure all abbreviations are defined at first use.
Major Questions for the Authors
Outcome Definition Justification: Why was a 90-day period selected as the cutoff for determining delayed initiation of anti-VEGF therapy? Is there clinical evidence supporting this threshold?
Case Definition: How were incident cases of diabetic macular edema (DME) differentiated from recurrent or chronic cases in the registry data?
Missing Data: How were missing values handled, particularly for sociodemographic variables such as race, income, and insurance status? Was multiple imputation used, or were these variables analyzed using complete case methods?
SES Measurement: Given that income was derived from ZIP code–level data, how do the authors account for the ecological fallacy risk in using area-based SES to infer individual-level disparities?
Insurance Interpretation: Can the authors provide additional interpretation or hypothesized mechanisms for the differences observed between Medicaid and Medicare coverage?
Clinical Relevance: What are the practical implications of a delayed VEGF start in this patient population? Could the authors elaborate on how this impacts long-term visual outcomes?
Conclusion
This study provides valuable insights into disparities in the initiation of VEGF therapy for DME using a rich data source. However, revisions are needed to strengthen the methodological transparency, clarify definitions, and contextualize findings more thoroughly. With these improvements, the manuscript could make a strong contribution to both ophthalmologic and health equity literature.

Author Response
Reviewer 2
General Assessment This manuscript addresses a clinically significant issue—the disparity in the initiation of VEGF inhibitor treatment for diabetic macular edema (DME), a key concern in ophthalmology and public health. The topic aligns well with the scope of the Journal of Clinical Medicine, particularly under the themes of health disparities, ophthalmology, and evidence-based treatment implementation. The structure is clear, and the authors present a solid rationale for their research question. However, there are multiple areas that require revision for clarity, methodological rigor, and scientific contribution.
Many thanks for the well-structured and generally very constructive criticism. We will be happy to take up your suggestions in order to improve clarity. Many thanks for the well-structured and generally very constructive criticism. We will be happy to take up your suggestions in order to improve clarity.
Strengths Timely and Relevant Topic: Addressing health disparities in DME treatment adds value to the literature and public health discussions. Real-World Data: The use of EHR-derived data from a large registry (IRIS) increases generalizability and reflects real clinical practice.
Statistical Rigor: The multivariable models and use of adjusted relative risks are appropriate and well-justified. Clear Presentation of Results: Tables and figures are informative and easy to interpret.
We also thank Reviewer 2 for the helpful comments to improve the manuscript. Unfortunately, some of the comments seem to contain relevant ‘hallucinations’, as they occur when using a LLM analysis. It was probably the wording of the title that caused the AI-generated system to make incorrect assumptions.
Weaknesses and Issues
Methodological Concerns
Inclusion Criteria: The authors excluded patients with a history of VEGF treatment, but it's unclear how recurrence vs. incident DME was handled. This should be more precisely defined.
The precise numbers of pretreated and treatment-naïve patients are reported. Therefore, it is not correct that patients with a history of VEGF treatment have been excluded.
Follow-Up Time: The choice of 90 days as the cutoff for treatment initiation is arbitrary and should be justified. Are there data showing that delays beyond 90 days lead to worse outcomes?
There was no cut-off of 90 days.
Missing Data: No mention is made about how missing data were handled (e.g., income, race/ethnicity). Clarify if imputation or complete case analysis was used.
This was a prospective, non-interventional trial. Loss of follow-up was discussed in the discussion.
Selection Bias: The paper should acknowledge possible biases in the subset of patients included in the IRIS registry and who had full variable documentation.
There was no relationship to the IRIS registry. Selection bias is already part of the limitations discussed (line 233).
Socioeconomic Data Limitations: Zip code–based income may not accurately reflect individual SES. This limitation should be discussed more directly.
Socioeconomic data was not part of this study, since the focus was on the phyiscians and the different treatment algorithms.
Interpretation of Results
The results suggest racial and insurance-based disparities, but causality should not be implied. The paper should be careful not to over-interpret associations.
The findings related to Medicare vs. Medicaid should be unpacked more fully—what are the clinical implications of this?
Although patients of three different European countries were recruited, there was no problem of racial or insurance-based disparities. Obviously, Medicare or Medicaid are not of relevance in these countries, as there is a universal health insurance and coverage of all relevant costs.
Clarity and Structure
Abstract: The abstract lacks precision. Specifically, “real-world practice patterns” and “sociodemographic factors” should be more specifically described.
If there are any clear suggestions how to add further parameters, we are open to rephrase the abstract. Both terms “real-world practice patterns” as well as “sociodemographic factors” are not part of the abstract.
Introduction: Well-motivated but overly broad. Focus more directly on gaps in knowledge about VEGF treatment disparities rather than general DME background.
Treatment disparities are not in the focus of the introduction. The first three abstracts of the article are just covering the lack in knowledge about the reasons of undertreatment. The different treatment algorithms are introduced.
Discussion: Needs a more in-depth comparison to prior studies. Several similar registry-based studies exist—how do these findings align or contrast?
It is of course important to provide a good overview of the existing literature. Nevertheless, no other prospective trial assessed the discrepancy between the regimen planned versus implemented.
Grammatical and Stylistic Issues
Passive Voice Overuse: Consider shifting to more active voice in several sections to enhance readability.
The language has been fundamentally revised. The use of passive voice was reduced.
Redundancy: Phrases such as “real-world treatment patterns in real-world data” should be avoided.
Although the expression “real-world” is used twice in the manuscript and once in the abstract, no such redundant combination is existent.
Inconsistency: Terms like “VEGF treatment” and “anti-VEGF therapy” are used interchangeably; standardize throughout the manuscript.
We searched the entire manuscript for the term “VEGF treatment”. However, we found only the term “anti-VEGF”. Therefore, there is no inconsistent use as supposed.
Recommendations for Major Revisions
Clarify Inclusion Criteria: Define incident cases of DME more clearly.
As written in line 84/85 (“Patients could be included in the study if they were being treated with ranibizumab 84 for any approved indication”) there was no further criterion of DME before included. The reviewer can find the precise characteristics of the patients with DME in Table A1.
Justify Outcome Definition: Explain why a 90-day window for treatment initiation was chosen.
There was no “90-day window”. Please check the precise description of the study protocol in the Methods section.
Improve Discussion of Limitations: Expand on data source limitations, potential confounders, and the nature of observational study bias.
All of the potential limitations of a non-interventional trial are already clearly specified (e.g. line 233). While there are potential confounders and potential selection bias, we do not agree on any data source limitations.
Refine Language: Polish the abstract, streamline the introduction, and use consistent terminology throughout.
The abstract was carefully revised. The introduction was
Enhance Clinical Implications: Add interpretation of what these disparities mean for clinicians and health policy.
Further points were added to the discussion
Minor Suggestions
Add full variable definitions in an appendix or supplemental table.
All abbreviations are clarified (lines 341ff.)
Include additional references to literature on racial/insurance disparities in ophthalmologic care.
Racial or insurance disparities are not the theme of this manuscript. However, we checked for the recent literature and added the relevant references.
Ensure all abbreviations are defined at first use.
The manuscript was carefully checked for any abbreviation.
Major Questions for the Authors
Outcome Definition Justification: Why was a 90-day period selected as the cutoff for determining delayed initiation of anti-VEGF therapy? Is there clinical evidence supporting this threshold?
There was no 90-day period selected. If the reviewer might find an additional
Case Definition: How were incident cases of diabetic macular edema (DME) differentiated from recurrent or chronic cases in the registry data?
The paper does not refer to any registry data. All patients, pretreated as well as treatment-naïve patients, signed an informed consent form.
Missing Data: How were missing values handled, particularly for sociodemographic variables such as race, income, and insurance status? Was multiple imputation used, or were these variables analyzed using complete case methods?
All relevant figures (Fig. 1 and Fig. 3) show the numbers of included patients. The manuscript reports on the number of discontinued patients rapidly increasing.
Information on income and race was not collected. All treatments are covered by the general health insurance.
SES Measurement: Given that income was derived from ZIP code–level data, how do the authors account for the ecological fallacy risk in using area-based SES to infer individual-level disparities?
It is not correct that income was derived from ZIP code-level data.
Insurance Interpretation: Can the authors provide additional interpretation or hypothesized mechanisms for the differences observed between Medicaid and Medicare coverage?
There was no coverage by Medicaid or Medicare. The study looked for the treatment numbers in a setting,
Clinical Relevance: What are the practical implications of a delayed VEGF start in this patient population? Could the authors elaborate on how this impacts long-term visual outcomes?
This article did not describe a delay in treatment, but an extensive and growing extent of undertreatment.
Conclusion
This study provides valuable insights into disparities in the initiation of VEGF therapy for DME using a rich data source. However, revisions are needed to strengthen the methodological transparency, clarify definitions, and contextualize findings more thoroughly. With these improvements, the manuscript could make a strong contribution to both ophthalmologic and health equity literature.
The reported difference refers to the regimen the physicians
Round 2
Reviewer 2 Report
Comments and Suggestions for Authors
Hermosillo, April 27th, 2025
From Plan to Practice: Disparities in VEGF Inhibitor Treatment Start for Diabetic Macular Edema
Dear Authors
After reading the authors' responses and comments, I would like to congratulate the authors for their good work and I agree that the manuscript should be accepted for publication.
